# Fatigue Life of Aluminum Alloys Based on Shear and Hydrostatic Strain

**DOI:** 10.3390/ma13214850

**Published:** 2020-10-29

**Authors:** Tadeusz Łagoda, Karolina Głowacka, Andrzej Kurek

**Affiliations:** Faculty of Mechanical Engineering, Opole University of Technology, 45-758 Opole, Poland; k.glowacka@po.edu.pl (K.G.); a.kurek@po.edu.pl (A.K.)

**Keywords:** multiaxial loading, strain, fatigue, multiaxial criteria, cyclic loading

## Abstract

The main purpose of this paper is to propose, based on the literature review, a new multiaxial fatigue strain criterion, analogous to the Dang Van stress criterion, considering the maximum amplitude of the shear strain and volumetric strain. The proposed strain criterion was successfully verified by fatigue tests in cyclic bending with torsion of specimens made of 2017A-T4 and 6082-T6 aluminum alloy. The scatter of test results for cyclic bending and the combination of cyclic bending and torsion is included in the scatter of tests for the cyclic torsion of the analyzed materials. Fracture surfaces for respective bending and torsion in the 6082-T6 aluminum test with strain control showed that, in the case of bending, cracks can be observed that develop from the surface of the specimen towards the bending plane. They are inclined from the fatigue crack at an angle of 45° in relation to the crack surface and the remaining cracks come from the static fracture. In the case of torsion, however, a conical fracture at 45° and a static torsion zone can be observed.

## 1. Introduction

From the analysis of the literature data [1], it can be seen that there is no single system to compare fatigue characteristics for tension-compression and torsion. This applies to both stress and strain characteristics; differences also exist between individual groups of materials, i.e., high- and low-alloy steels, aluminum alloys, non-ferrous metals etc. An additional problem is the consideration of stress and strain gradients in case of bending and torsion. It is noteworthy that, in this case, the authors usually do not take into account plastic strain and treat the material as elastic-brittle.

This paper presents the result of fatigue tests of 2017A-T4 and 6082-T6 aluminum alloys under different load conditions, namely alternating bending and bilateral torsion and combinations of these load conditions. The phenomenon of tension and bending occurs in virtually every industry [2], so it is not surprising that these two load states are also considered in relation to material fatigue [3,4]. Most of today’s fatigue characteristics are performed under tensile and compressive conditions. Unfortunately, such a loading state very rarely occurs in real mechanical structures subjected to fatigue loads [5]. More often it is oscillatory bending [6]. Such a situation makes the relation between the tension-compression and alternating bending fatigue characteristics an interesting and current topic of considerations [7]. It should be noted here that in the case of bending, these characteristics are most often constructed with the adoption of an ideally elastic body model. As already mentioned, for the analyzed materials, within a large fatigue strength range, the tested materials behave as ideally elastic [8]. In order to perform such an analysis, it is necessary to assume a linear distribution of strains across the section, the Ramberg–Osgood relationship at each point and the condition that the integral of stresses across the section balances the given bending moment. On the basis of previous tests of aluminum alloy 6082-T6 [8], it was possible to conclude that the stress and strain gradient remains unaffected by the fatigue life of the tested material because the fatigue characteristics for alternating bending and tension-compression were practically identical. Earlier analyses of these alloys also showed that plastic deformations are virtually non-existent and the material behaves like an elastic-brittle material in a wide range [8].

In the literature on fatigue life assessment, three types of multiaxial fatigue criteria (due to their components) reducing complex states to a suitable uniaxial one can be noticed. These are the best known and popular stress models, less common strain models and the so-called energy models (otherwise stress–strain models). Recently, the most popular are those criteria that are defined in the critical plane. Then, the equivalent stress value is usually a sum of shear and normal stress with weighting factors. Another proposal is to add shear and hydrostatic stress with successive weighting factors. In this respect, the best known and developed model is the Dang Van’s proposal.

The Dang Van criterion [9,10] is distinguished by a mesoscopic (grain level) scale of stress observation. The Dang Van criterion assumes that fatigue does not occur if all grains reach the elastic shakedown state. This means that after the initial loading period, the material will be isotropically hardened and the further relation between stress and strain will take place the elastic range. On a macroscopic scale, the material may be in the elastic state and in this case two states may be distinguished:All grains have reached the elastic deformation state, which means unlimited fatigue life;Some grains have preferential slip planes oriented in a way that they do not reach the stable state of elastic deformation, but they change to the stable plastic deformation (plastic shakedown) state or the unstable plastic deformation (ratcheting) state, which means damage cumulation and limited fatigue life.

The Dang Van criterion deletes the crack initiation condition and does not allow the fatigue life calculation. The condition of exceeding the stabilized elastic deformation state is dependent on mesoscopic shear and volumetric stresses. These stresses are related by a linear function in the form of
(1)$$\tau \left(t\right)+a\sigma_{h}\left(t\right)\leq b.$$

In amplitudes this criterion can be formulated as
(2)$$\tau_{a,max}+a\sigma_{h}\leq b$$
where a, b are constants determined from uniaxial fatigue tests
(3)$$a=\left(\tau_{af}-0.5\sigma_{af}\right)/\left(\sigma_{af}/3\right)$$
$$b=\tau_{af},$$

$\tau_{a,max}$ —the value of the maximum mesoscopic shear stress is calculated from the mesoscopic principal stresses according to the Tresca hypothesis, and the hydrostatic stress is calculated as
(4)$$\sigma_{h}=\frac{\sigma_{1}+\sigma_{2}+\sigma_{3}}{3}.$$

This criterion is very popular in French research centers, with the vast majority of researchers using this criterion, assuming that the stresses determined by standard macroscopic scale methods are proportional to those on the mesoscopic scale, which allows the use of the Dang Van criterion in engineering calculations [11].

The main reason of this paper is proposes, a new multiaxial fatigue strain criterion, analogous to the Dang Van stress criterion, considering the maximum amplitude of the shear strain and volumetric strain-based 2017A-T4 and 6082-T6 aluminum alloys under different load conditions.

## 2. Fatigue Strain Fracture Criteria

When analyzing the issue of tension-compression, the Manson–Coffin–Basquin (MCB) model must be mentioned [12,13,14,15]:(5)$$\varepsilon_{a,t}=\varepsilon_{a,e}+\varepsilon_{a,p}=\frac{{\sigma^{\prime }}_{f}}{E}{\left(2N_{f}\right)}^{b}+{\varepsilon^{\prime }}_{f}{\left(2N_{f}\right)}^{c}$$
where:

$\varepsilon_{a,t}$—total strain amplitude expressed as the sum of amplitudes of elastic strain $\varepsilon_{a,e}$ and plastic strain, $\varepsilon_{a,p}$

$2N_{f}$—number of load recurrence (semi-cycles),

*E*—Young’s modulus,

${\sigma^{\prime }}_{f}$, *b*—coefficient and exponent of fatigue strength,

${\varepsilon^{\prime }}_{f}$, *c*—coefficient and exponent of fatigue plastic deformation.

The original MCB characteristic was designed for tension-compression analysis of strain, stress and number of cycles to failure.

Equation (5) is used only if it is possible to determine separately both the elastic $\varepsilon_{ae}$ and plastic $\varepsilon_{ap}$ total strain component $\varepsilon_{at}$ [16,17].

Then for cyclical loads we obtain:(6)$$\varepsilon_{a,e}={\frac{\sigma_{a}}{E}}^{\;}$$
and (7)$$\varepsilon_{a,p}=\varepsilon_{a,t}-\varepsilon_{a,e}$$


This relationship is described in the Ramberg–Osgood equation [18]:(8)$$\varepsilon_{a,t}=\varepsilon_{a,e}+\varepsilon_{a,p}=\frac{\sigma_{a}}{E}+{\left(\frac{\sigma_{a}}{K^{\prime }}\right)}^{\frac{1}{n^{\prime }}}$$
where:

$\sigma_{a}$ —stress amplitude,

$K^{\prime }$ —cyclic strength factor,

$n^{\prime }$ —exponent of cyclic hardening.

Similar to the Manson–Coffin–Basquin (MCB) Equation (5), a similar shearing model can be proposed for tension-compression [19].
(9)$$\gamma_{a,t}=\gamma_{a,e}+\gamma_{a,p}=\frac{{\tau^{\prime }}_{f}}{G}{\left(2N_{f}\right)}^{b_{0}}+{\gamma^{\prime }}_{f}{\left(2N_{f}\right)}^{c_{0}}$$
where:

$\gamma_{a,t}$ —total non-dilatational strain amplitude expressed as the sum of the amplitudes of the pure elastic strain $\gamma_{a,e}$ and plastic strain $\gamma_{a,p}$,

$2N_{f}$—number of load recurrence (semi-cycles),

G —shear modulus,

${\tau^{\prime }}_{f}$, $b_{0}$ —coefficient and exponent of shear fatigue strength,

${\gamma^{\prime }}_{f}$, $c_{0}$ —coefficient and exponent of fatigue plastic deformation for shear.

Equation (9) as well as Equation (8) for tension-compression is only applied if it is possible to determine separately both the elastic $\gamma_{a,e}$ and the plastic $\gamma_{a,p}$ component of the total strain $\gamma_{a,t}$.

Then for cyclical loads we obtain: (10)$$\gamma_{a,e}=\frac{\tau_{a}}{G}$$
and
(11)$$\gamma_{a,p}=\gamma_{a,t}-\gamma_{a,e}$$

Most structural analyses are carried out using stress and strain tensors. An exception to this rule is the model based on the maximum shear stress amplitude and hydrostatic stress—Equation (2). What is interesting is whether the same approach can be applied to the strain model.

Therefore, a mean value can be determined; in other words, the hydrostatic strain, on the basis of the normal components of the strain tensor given in the general formula
(12)Tε=[εxxγxy2γxz2γyx2εyyγyz2γzy2γzy2εzz]
i.e., by analogy to Equation (4) we get
(13)$$\varepsilon_{h}=\frac{\varepsilon_{xx}+\varepsilon_{yy}+\varepsilon_{zz}}{3}$$

For further analysis, let us consider simple load conditions. For pure torsion at a given moment of time we have a strain tensor
(14)Tε=[0γxy20γyx200000]

In this case, the deviator is equal to the tensor because in this system, the hydrostatic deformation is equal to 0.

In such a case, the maximum amplitude of both the deviator and tensor of shear strain is
(15)(γa2)max= γyxa2

Another simple case of load to consider is pure tension (compression), then in a given moment of time we have a strain tensor
(16)$$T_{\varepsilon }=\left[\begin{array}{ccc} \varepsilon_{xx} & 0 & 0\\ 0 & -\vartheta \varepsilon_{xx} & 0\\ 0 & 0 & -\vartheta \varepsilon_{xx} \end{array}\right]$$
and the mean value (hydrostatic strain) according to Equation (13) is expressed as
(17)$$\varepsilon_{h}=\frac{\varepsilon_{xx}-\vartheta \varepsilon_{xx}-\vartheta \varepsilon_{xx}}{3}=\frac{1-2\vartheta }{3}\varepsilon_{xx}$$

Here, it is worth noting that in the case of a strain tensor for uniaxial tension-compression we have
(18)(γa2)max

As mentioned earlier in the literature, we most often deal with multiaxial fatigue criteria formulated in terms of stress. However, in practice, we measure strain because strains are measurable, while the stress criteria fail when plastic deformations occur, and it is worth remembering that the strain criteria are more universal in the full range of fatigue life, although they are more difficult in formulation. Therefore, below you will find the strain criteria that are most commonly used in practice.

There are few literature reviews of strain criteria with particular emphasis on those defined in the critical plane. One of the exceptions is the work of Karolczuk [20,21].

Brown and Miller [22] proposed the criterion of multiaxial material fatigue, which assumes that fatigue life is a non-linear function of the strain state. The contour with a constant strength is presented in the general equation as
(19)$$\frac{\varepsilon_{1}-\varepsilon_{3}}{2}=f\left[\frac{\varepsilon_{1}+\varepsilon_{3}}{2}\right]\;\mathrm{or}\;\frac{\gamma_{13}}{2}=f\left[\varepsilon_{n}\right].$$

The criterion was proposed as a result of observation of initiation and propagation of fatigue cracks. It was observed that in many metals, the crack initiation process occurs on crystallographic slip planes and is controlled by shear strain. In the first stage (Stage I), cracks propagate at the maximum shear planes as a process of slip and decohesion. For most materials, the second stage of propagation (Stage II) is a continuation of slip and decohesion processes. However, cracks may propagate through voids (defects), especially in brittle materials (grey cast iron). The second but important effect is the effect of normal strain acting in the plane of the maximum shear strain. This strain affects the mobility of dislocation and decohesion associated with slip processes. Considering both effects, Brown and Miller came to the conclusion that the maximum shear and normal strain in the plane of maximum non-dilatational strain are the parameters that determine fatigue life. The critical plane is the plane with the greatest shear strain $\gamma_{13}$. Brown and Miller have introduced two types of cracks: type A and B. For the case A, the crack develops along the surface of the material. For the case B, the crack develops inside the material. Cases A and B occur for both stage I and II. For pure torsion the case A applies, while the case B is for biaxial tension-compression (e.g., cross specimens). For a tension-torsion combination, there is always the case A. This criterion, formulated in its general Equation (19), was reduced to a linear function by Kandil–Brown–Miller [23].
(20)$$\frac{\Delta \gamma_{13}}{2}+S\Delta \varepsilon_{n}=C,$$
where *S* is an experimentally determined constant called the effect factor of normal strain. It should be noted here that the use of deformation ranges in Equation (20) is justified only in the case of proportional loads. In order to take into account the effects of disproportionality and varying amplitude of strains, Wang and Brown [24] proposed modifications to Criterion (20) in the form of
(21)$$\frac{\Delta \gamma_{ns}}{2}+S\Delta \varepsilon_{n}^{*}=\left(1+\nu_{e}+\left(1-\nu_{e}\right)S\right)\frac{\sigma_{f^{\prime }}}{E}{\left(2N_{f}\right)}^{b}+\left(1+\nu_{p}+\left(1-\nu_{p}\right)S\right)\varepsilon_{f^{\prime }}{\left(2N_{f}\right)}^{c}.$$

The difference between Equations (20) and (21) lies in the different definitions of the range of normal strain. Normal strain $\varepsilon_{n}^{*}$ (normal strain excursion) is calculated on the plane of the maximum range of the shear strain in the cycle with the greatest range $\Delta \gamma_{ns}$. The range of non-dilatational strain is calculated along the direction s in the plane with normal n:(22)$$\varepsilon_{n}^{*}=\underset{t_{A}<t<t_{B}}{max}\left\{\varepsilon_{n}\left(t\right)\right\}-\underset{t_{A}<t<t_{B}}{min}\left\{\varepsilon_{n}\left(t\right)\right\}=\varepsilon_{n,\;max}-\varepsilon_{n,\;min},$$
where: $t_{A}$ and $t_{B}$ are the beginning and the end of the load cycle. The value $\varepsilon_{n}^{*}$ is always taken as a positive value even in compression, because for a range with a small number of cycles (LCF) the influence of the mean strain value is small (according to the authors [24]). The critical plane is one of the planes with the maximum range of the non-dilatational strain, for which $\Delta \varepsilon_{n}^{*}$ it assumes a higher value. The authors signal the usefulness of the proposed criterion for a range of high number of cycles (HCF) provided the influence of the average load value in Equation (20) is considered. Equation (21) covers both A and B cracking cases:(23)$$\gamma_{eq,a}=\frac{\Delta \gamma_{ns}}{2}+S\Delta \varepsilon_{n}-\mathrm{case}\;A,$$
(24)$$\gamma_{eq,a}=\frac{\Delta \gamma_{ns}}{2}-\mathrm{case}\;B.$$

For case B, no impact of load disproportionality is assumed (*S* = 0). In case B, the plane of maximum shear strain is at an angle of $45^{\circ }$ to the surface of the material.

Socie et al. [25,26] and Liu et al. [27] observed fatigue cracks in the field of deformation. According to them, the normal strain $\varepsilon_{n}$ in the plane of maximum shear strain $\gamma_{ns}$ accelerates the material degradation process by opening the fatigue gap, thus reducing the friction forces between the slip planes. The criterion also takes the effect of the mean normal stress value $\sigma_{n,m}$ in the plane with maximum amplitude of non-dilatational strain $\gamma_{ns,a}$ into account. Equivalent strain amplitude $\gamma_{eq,a}$ is calculated from the relationship
(25)$$\gamma_{eq,a}=\gamma_{ns,a}+\varepsilon_{n,a}+\frac{\sigma_{n,m}}{E}$$

The material will not be subject to failure before the number of cycles N satisfy the following condition:(26)$$\gamma_{eq,a}\leq \gamma_{kr}\left(N\right),$$
where $\;\gamma_{kr}\left(N\right)$ is a critical strain dependent on the number of cycles $N=N_{f}$. Based on the fatigue analysis of various materials, Fatemi and Socie [28] noted that the Equation (25) does not take into account the hardening of the material occurring during non-proportional loading. In order to take this phenomenon into account, they modified Equation (25), replacing the normal strain value $\varepsilon_{n,a}$ in the critical plane, with the maximum value of normal stress $\sigma_{n,\;max}$. The critical plane is the plane with the maximum amplitude of shear strain $\gamma_{ns,a}$. For a given number of cycles up to the failure $N_{f}$, the aforementioned strain and stress form the function
(27)$$\gamma_{ns,a}\left(1+n\frac{\sigma_{n,\;max}}{R_{e}}\right)=const,$$
where n is an experimentally selected constant. For the low cycle load range (LCF), the relation (4.53) is expressed as a function of the number of cycles to failure as [28]
(28)$$\begin{array}{c} \gamma_{ns,a}\left(1+n\frac{\sigma_{n,\max\;}}{R_{e}}\right)=\left(1+\nu_{e}\right)\frac{\sigma_{f^{\prime }}}{E}{\left(2N_{f}\right)}^{b}+\frac{n}{2}\left(1+\nu_{e}\right)\frac{\sigma_{f^{\prime }}^{2}}{ER_{e}}{\left(2N_{f}\right)}^{2b}+\\ \left(1+\nu_{p}\right)\varepsilon_{f^{\prime }}{\left(2N_{f}\right)}^{c}\frac{n}{2}\left(1+\nu_{p}\right)\frac{\varepsilon_{f^{\prime }}\sigma_{f^{\prime }}}{R_{e}}{\left(2N_{f}\right)}^{b+c} \end{array}$$

Macha, similarly to the stress criterion [29], made a generalization of some of the strain criteria. The detailed assumptions of this criterion are as follows:-Fatigue cracking occurs under normal strain $\varepsilon_{n}\left(t\right)$ and shear strain $\varepsilon_{ns}\left(t\right)$ occurring in the direction s on the critical plane of normal n;-The direction s in the plane of normal n coincides with the mean direction of the maximum non-dilatational strain;-For a given fatigue life, the maximum value of the linear strain combination $\varepsilon_{n}\left(t\right)$ and $\varepsilon_{ns}\left(t\right)$ under multiaxial random load conditions satisfies the equation

(29)$$\underset{t}{max}\left\{b\varepsilon_{ns}\left(t\right)+k\varepsilon_{n}\left(t\right)\right\}=q$$
where b, k, q are constants to choose the specific version of the criterion.

Ogonowski and Łagoda [30] detailed Macha’s general criterion [29], giving the formula for determining the b and k coefficients. For the plane of maximum non-dilatational strains, the equivalent strain is determined from the following general equation
(30)$$\varepsilon_{eq}\left(t\right)=\underset{ns}{max}\left\{b\varepsilon_{ns}\left(t\right)\right\}+k\varepsilon_{n}\left(t\right)$$
where ns is the critical plane with maximum shear strain. In order to apply the criterion to the low number of cycles (LCF) range, Karolczuk, Łagoda, Ogonowski proposed the division of strain into elastic and plastic [31]. This procedure allowed to derive the equations into the form of coefficients b and k based on uniaxial fatigue tests. Therefore, Equation (30) takes the form of
(31)$$\varepsilon_{eq}\left(t\right)=\underset{ns}{max}\left\{b\varepsilon_{ns}\left(t\right)\right\}+k\varepsilon_{n}\left(t\right)$$
where (32)$$b=2\varepsilon_{a}\left(2N_{f}\right)/\gamma_{a}\left(2N_{f}\right)$$
is the coefficient depending on the number of cycles to failure, $N_{f}$; $k_{e}$, $k_{p}$ are the coefficients depending on the number of cycles $N_{f}$ for the elastic and the plastic parts, respectively, derived from the cyclic torsion test; $\varepsilon_{n}^{e}$, $\varepsilon_{n}^{p}$ is the adequately elastic and plastic part of the normal strain in the plane of normal n. The $k_{e}$, $k_{p}$ factors are determined as follows:(33)$$k_{e}=\frac{2}{1-\nu_{e}}\left[1-\frac{\varepsilon_{a}^{e}\left(2N_{f}\right)}{\gamma_{a}^{e}\left(2N_{f}\right)}\left(1+\nu_{e}\right)\right]$$
(34)$$k_{p}=\frac{2}{1-\nu_{p}}\left[1-\frac{\varepsilon_{a}^{p}\left(2N_{f}\right)}{\gamma_{a}^{p}\left(2N_{f}\right)}\left(1+\nu_{p}\right)\right]$$
where $\nu_{e}$, $\nu_{p}$ are Poisson’s ratios for elastic and plastic strain respectively ($\nu_{p}=0.5\;and\;\nu_{p}=0.5$); $\gamma_{a}^{e}$, $\gamma_{a}^{p}$ is the correspondingly elastic and plastic part of the shear strain in the cyclic torsion test. Factors b and $k_{e}$, $k_{p}$ depend on the number of cycles to failure $N_{f}$, which requires the method of finding a solution to a non-linear equation with one unknown ($N_{f}$). This equation results from the calculation of the equivalent strain amplitude $\varepsilon_{eq,a}$ in the critical plane (pre-determined plane of maximum strain) and comparison of this value with the strain amplitude of the Manson–Coffin–Basquin equation.

Shang De-Guang and Wang De-Jun [32,33] proposed a criterion for a small number of cycles in a non-linear form based on Huber–Mises–Hencky’s (H–M–H) proposal. The equivalent form of strain amplitude is calculated from the formula
(35)$$\varepsilon_{eq,a}=\sqrt{{\left(\varepsilon_{n}^{*}\right)}^{2}+\frac{1}{3}\gamma_{ns,a}^{2}}$$
where $\varepsilon_{n}^{*}$ is the normal strain in the critical plane calculated similar to Brown–Miller–Kandil’s proposal [23] Equation (20). The normal and shear strains are determined in the critical plane with the maximum shear strain. In the case of several planes with the same maximum shear strain value, the critical plane is the plane with the greater value of normal strain $\varepsilon_{n}^{*}$. The number of cycles to failure is determined from the Manson–Coffin–Basquin fatigue characteristics. The direct transfer of the mathematical formula of the H–M–H hypothesis to calculate the equivalent strain amplitude in the critical plane has no physical justification and raises a number of doubts concerning the correctness of such a transfer. There is no connection between the energy of the non-dilatational strain, which is the basis of the H–M–H hypothesis, and the proposed solution.

## 3. Materials and Methods

In the present study, the tension-compression and alternating bending characteristics and bilateral torsion characteristic with the assumption of an ideally elastic body model were compared using stress and strain characterization models [34,35,36]. The analysis was carried out on the basis of fatigue tests of aluminum alloy 2017A and 6082 in the two load states under consideration. Moreover, the analysis was carried out on the basis of experimental studies with a proportional combination of cyclic bending with torsion.

The chemical composition of the material is summarized in Table 1. While the basic mechanical properties of the materials under consideration are presented in Table 2. The main difference between these materials is that 2017A aluminum is rich in copper, and 6082 aluminum has an increased silicon content compared to 2017A.

Basic research within a small number of tension-compression cycles was carried out for 2017A-T4 at the Opole University of Technology [37], and for 6082-T6 in cooperation with the Institute Laboratory for Materials and Construction Research at the University of Science and Technology in Bydgoszcz [38].

Fatigue tests for alternating bending, bilateral torsion and combinations of cyclic proportional bending and torsion were carried out on the fatigue machines, which in the equipment of the laboratory of the Department of Mechanics and Machine Design of the Opole University of Technology [39].

### 3.1. Tension and Compression Tests

The aim of the tests was to determine the basic fatigue characteristics of aluminum alloy specimens at ambient temperature.

After the analysis of the results of the static tension test, it was proposed to carry out the low cycle tests for aluminum 2017A-T4 at six levels of total strain amplitude $\varepsilon_{ac}$: 0.3%, 0.4%, 0.5%, 0.6%, 0.65%, 0.7% with strain ratio R = −1.13 samples were used for the tests.

For 6082-T6 aluminum, on the other hand, there are five levels of total strain amplitude $\varepsilon_{ac}$: 0.35%, 0.5%, 0.8%, 1.0%, 2.0% with strain ratio R = −1.16 samples were used.

The specimens were made according to the Polish standard PN-84/H-04334 [40].

In order to determine the levels of strain under low-cycle test conditions, they were preceded by a static tensile test. During the test, specimens of 2017A and 6082 alloys were used for fatigue test, the forms of which are shown in Figure 1a,b, respectively.

Based on the results of the uniaxial tension-compression fatigue tests, the material constants found in the Manson–Coffin–Basquin Equation (5) and Ramberg–Osgood Equation (8) characteristics are listed in Table 3.

### 3.2. Cyclic Proportional Torsion and Bending Tests

Cylindrical “diabolo” type specimens without geometric notch were used in fatigue tests. The geometry of the specimens used results from the facilitated location of the site with the highest stress. Fatigue tests were carried out on “diabolo” specimens (Figure 2) at the test stands belonging to the Faculty of Mechanical Engineering of the Opole University of Technology. The starting material was a circular bar with a diameter of φ16 mm. The tests for both materials were performed for a combination of bending and torsion at controlled torque. In addition, for aluminum 6082, controlled strain tests were performed for alternating bending and bilateral torsion.

Fatigue tests at the controlled moment were performed on the stand shown in Figure 3, the strain tests were performed on the new stand shown in Figure 4. In this case, the amplitude of lever deflection was controlled, which, in effect, gives control of the strain on the specimen. A detailed description of the operation of these stands can be found in numerous works of the Opole University of Technology employees, especially in [39].

Using both the torque control station and strain control station, and by means of an appropriate head positioning (Figure 5), various combinations of proportional cyclic bending and torsion were achieved.

The value of the torsion $M_{s}\left(t\right)$ and bending $M_{g}\left(t\right)$ moments is linked to the relationship
(36)$$tg\beta =\frac{M_{s}\left(t\right)}{M_{g}\left(t\right)}.$$

If β=0 the specimen is bent, at β=π/2 the specimen is twisted.

In intermediate positions 0<β<π/2, both moments occur simultaneously according to dependencies:(37)$$M_{g\alpha }\left(t\right)=\left(M_{a}sin\omega t\right)cos\beta ,$$
(38)$$M_{s\alpha }\left(t\right)=\left(M_{a}sin\omega t\right)sin\beta .$$

The result of both moments is a stress state in which the stresses σ(t) and τ(t) change their values according to phase and frequency (proportional loads):(39)$$\sigma_{\beta_{a}}\left(t\right)=\sigma_{a\alpha }sin\omega t,$$
(40)$$\tau_{\beta_{a}}\left(t\right)=\tau_{a\alpha }sin\omega t,$$

The values of normal stresses $\sigma_{\beta_{a}}\left(t\right)$ and shear stresses $\tau_{\beta_{a}}\left(t\right)$ within the elastic range can be determined
(41)$$\sigma_{\beta }\left(t\right)=\frac{M_{g\alpha }\left(t\right)}{W_{x}},$$
(42)$$\tau_{\beta }\left(t\right)=\frac{M_{s\alpha }\left(t\right)}{W_{0}},$$
where
(43)$$W_{x}=\frac{\pi d^{3}}{32}$$
is the cross-section bending modulus, and
(44)$$W_{0}=\frac{\pi d^{3}}{16}$$
is the cross-section twisting modulus.

For both materials, tests were carried out with the following head settings, i.e., angle

β=0° (bending), β=22.5°, β=45°, β=67.5° and β=90° (torsion). The intermediate angle setting gives the ratio of amplitudes for non-dilatational strain
(45)$$k_{strain}=\frac{\gamma_{a}/2}{\varepsilon_{a}}$$

The result is a combination of bending and torsion $k_{strain}$: 0.318, 0.660 and 1.352.

For both controlled moment and controlled strain fatigue tests, the fatigue life was assumed to be the point when the crack was visible to the naked eye, i.e., about 1 mm. In the case of controlled strain, the course of the moment amplitude was recorded simultaneously, which, at the same time, fell rapidly. In these tests, the strain in bending at the most stressed point and the shear strain for torsion were controlled accordingly. The pattern of moment was recorded in parallel. The moment of initiation of the fatigue crack, and thus the fatigue life, was assumed to be the moment of rapid (15%) decrease, which was equivalent to the appearance of a crack visible to the naked eye on the surface of the tested specimen.

## 4. Results

Figure 6 and Figure 7 show microscopic images of fracture surfaces for bending and torsion for 6082-T6 aluminum tests, respectively. These tests were carried out with controlled strain, which resulted in a very stable operation of the new stand while maintaining the structure of the fracture during its cracking. In the case of bending, cracks can be observed which develop from the surface of the specimen towards the bending plane. They are inclined at 45° to the crack surface. The development of such cracks has already been pointed out in [41]. The cross-section photographs and 3D scans of fatigue cracks were taken on two different optical scanning microscopes; for Figure 6 and Figure 7, we used Huvitz HRM-300 and for 8 and 9 Sensofar S neox3D optical profiler with focus variation method.

In case of aluminum, a neutral plain also passes through, approximately, the geometrical center of the specimen; however, it is not as clear as in case of the analyzed steel. Yet, in this case, there are clear pits oriented at the angle of 45° towards the breakthrough surface. Such a cracking manner is described by, inter alia, Schijve in [42].

However, in case of the tests analyzed, these pits are of a slightly different nature. The pits visible in Figure 8 are in the shape of double shear lips and they run from the external surface to approximately 2/3 length along the line perpendicular to the bending plane. This cracking manner was anticipated and described in the study [43]. These pits are better visible in a microscopic image.

Additionally, a typical cross-section of the sample for alternating bending is shown in Figure 8 and for torsion in Figure 9. These images clearly show the point of initiation and the neutral plane with respect to which the bending took place, as well as the part of the static fracture and, in the case of torsion, the conical crack at an angle of 45° and the zone of static torsion.

## 5. Discussion of Comparison of Strain Fatigue Characteristics

The comparison of strain fatigue characteristics is presented in more detail, among others, in [44]. As it has been shown, the relationship between the characteristics, and in particular between individual constants occurring in the tension-compression (oscillatory bending) and shear (bilateral torsion) characteristics, varies and depends largely on the type of material. Sometimes fatigue is determined by normal strain, sometimes by shear strain and sometimes by a combination of these two. As mentioned earlier, the full fatigue characteristics according to Equations (5) and (9) will not be used in this work because the aluminum alloys analyzed behave like practically elastic bodies in a wide range. Previous [45,46] analyses for these materials show that even at maximum loads in the analyzed tests the assumption of an elastic-plastic body significantly hinders the modelling of fatigue strength and the decrease in stress amplitude is only 2 MPa, which corresponds to very slight plastic strain and, consequently, a minimal increase in total strain. Therefore, simplified formulas analogous to Basquin’s model were used for normal strain for oscillatory pendulum bending:(46)$$\varepsilon_{a}={\varepsilon^{\prime \prime }}_{f}\cdot N_{f}^{b_{\varepsilon }}$$
where
(47)$${\varepsilon^{\prime \prime }}_{f}={\sigma^{\prime }}_{f}/E\cdot$$
and the shear strain for bilateral torsion:(48)$$\gamma_{a}={\gamma^{\prime \prime }}_{f}/2\cdot N_{f}^{b_{\gamma }}$$
where
(49)$${\gamma^{\prime \prime }}_{f}=\frac{{\tau^{\prime }}_{f}}{G}.$$

The results of the calculations of fatigue characteristics as a function of the number of cycles according to Equations (46) and (47) are presented in Table 4, and graphically in Figure 10a and Figure 11a for aluminum 2017A and 6082, respectively. Such a procedure could be performed because, in a very large range of fatigue lifetime, the analyzed materials behave as perfectly elastic materials. The necessary constants were determined from the tension-compression and torsion relationships for the durability in the middle durability range, i.e., 10^5^ cycles. This could be done this way because although these characteristics are not parallel, they do not differ significantly from this parallelism (0.118/1.208 and 0.213/0.188 for 2017A and 6082, respectively). The oscillatory bending characteristics and bilateral torsion are almost parallel, which is very important [46], but the individual calculations in the last two columns of the table mentioned above were made for fatigue strength from about the middle part of the determined characteristics, that is, for 105 cycles.

Earlier analyses [47] showed that the ratio ${\tau^{\prime }}_{f}/{\sigma^{\prime }}_{f}$ theoretically fits in the range of
(50)$$\frac{1}{\sqrt{3}}<\frac{{\tau^{\prime }}_{f}}{{\sigma^{\prime }}_{f}}<\frac{1}{1+\vartheta }$$
but for different materials, the ratio often goes beyond that.

With the assumption of elasticity, Equation (50) a can be presented in the strain-related form as:(51)$$\frac{1+\vartheta }{\sqrt{3}}<\frac{\gamma_{af}/2}{\varepsilon_{af}}<1$$

Using data in Table 4, the relation from the equation can also be determined for 105 cycles. As a result, it turns out that this ratio is within the range given in Equation (50) for aluminum 6082-T6 and is slightly higher than 1 for 2017A. However, it can be seen that if the ratio of the newly introduced values $\frac{{\gamma^{\prime \prime }}_{f}/2}{{\varepsilon^{\prime \prime }}_{f}}$ was used, we would have a perfect match of results with the theory.

Further analysis shows that these characteristics are almost parallel and their coefficients are very similar. Unfortunately, the lower fatigue strength range for which pure twisting tests were performed limits the possibility of a full analysis of these relationships. However, it can be clearly seen that for a smaller fatigue strength, the scatter of test results is small, and for larger ones it is relatively high.

It can be noted that many models (19)–(21), (23)–(25), (27), (28), (30), (31), (35) are based on the amplitude of the maximum shear strain. Therefore, such amplitudes were determined for alternating bending tests and compared with the shear strain amplitudes for aluminum 2017A and 6082 respectively in Figure 10b and Figure 11b. A simple analysis shows that for aluminum 6082, the maximum amplitude of the shear strain is a sufficient damage parameter, and $\frac{\gamma_{af}/2}{\varepsilon_{af}}$(10^5^) is around 1. This cannot be said for the second aluminum 2017A.

Therefore, a new form of strain criterion was proposed, based on maximum amplitude of the shear and hydrostatic strain, analogous to the stress Equation (2). This criterion is as follows
(52)$$\gamma_{a,max}/2+a_{\varepsilon }\varepsilon_{h}\leq b_{\epsilon }$$

Therefore, the expression for equivalent strain amplitude can be formulated as
(53)$$\gamma_{a,eq}/2=\gamma_{a,max}/2+a_{\varepsilon }\varepsilon_{h}$$

Taking into account simple load conditions and using the data presented in Table 4, the specific form of this criterion is obtained from the calculations for the number of cycles to destruction 105, i.e., for $\frac{\gamma_{af}/2}{\varepsilon_{af}}$
(54)aε=31−2ϑ(γaf2εaf)−3(1+ϑ)2(1−2ϑ)
(55)$$b_{\varepsilon }=\frac{\gamma_{af}}{2}$$
where $\varepsilon_{af}$ and $\gamma_{af}\;$ concern the amplitudes of the relevant normal strain and shear strain from the strain fatigue characteristics for the specified fatigue strength. In the case under consideration, this is assumed for a life of 10^5^ cycles.

Graphic interpretation of this criterion is shown in Figure 12. This figure shows the general position of the characteristic Equation (53). Basically, the ratio $\frac{\gamma_{af}/2}{\varepsilon_{af}}$ must not be less than 1 and the corresponding k-factor less than 0. If that were the case, the characteristics would increase and it would mean that hydrostatic strain increases the fatigue strength. Therefore, in Table 4 for aluminum 6082-T6 extreme values of 1 and 0, respectively, are adopted. This discrepancy must result from the scattering of the fatigue life and, therefore, the inaccurate determination of these coefficients.

Then, using the individual test results for alternating bending, bilateral torsion and a combination of these loads at a different $k_{strain}$ ratio Equation (45) of the bending and torsion combination for the two aluminum alloys under analysis. A new form of criterion Equation (53) was used for the calculations, and the results are shown in Figure 13 and Figure 14 against the background of the fatigue characteristics for pure cyclic torsion, extending these characteristics with a dashed line over the range of lower fatigue strength. From the analysis of these figures, it can be seen that the strain criterion was successfully verified by fatigue tests in cyclic bending with torsion of samples made of alluvium alloy 2017A-T4 and 6082-T6. It can be seen that the scatter of the test results for cyclic bending and the combination of cyclic bending and torsion is included in the scatter of the cyclic torsion tests on the materials analyzed.

## 6. Conclusions

The paper proposes a new strain criterion of multiaxial fatigue analogous to the Dang Van stress criterion, taking into account the maximum amplitude of the shear strain and hydrostatic strain.The proposed strain criterion was successfully verified by fatigue tests in cyclic bending with torsion of specimens made of 2017A-T4 and 6082-T6 aluminum alloy.The scatter of test results for cyclic bending and the combination of cyclic bending and torsion are included in the scatter of tests for cyclic torsion of the analyzed materials.The fracture surfaces for bending and torsion, respectively, in 6082-T6 aluminum test with strain control showed that in bending, cracks can be observed which develop from the surface of the test piece towards the bending plane. They are inclined from the fatigue crack at an angle of 45° in relation to the crack surface and then part of the static fracture. In the case of torsion, a conical fracture at 45° and a static fracture zone can be observed.

## Figures and Tables

**Figure 1 materials-13-04850-f001:**
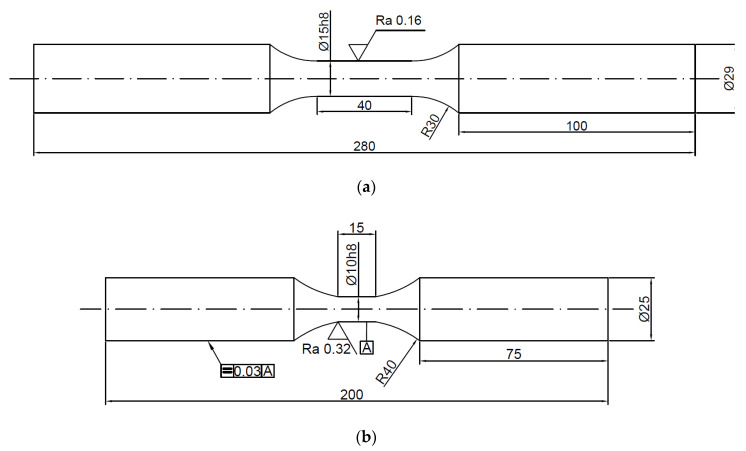
Uniaxial tension/compression specimens (**a**) 2017A, (**b**) 6082, (dimensions are in mm).

**Figure 2 materials-13-04850-f002:**
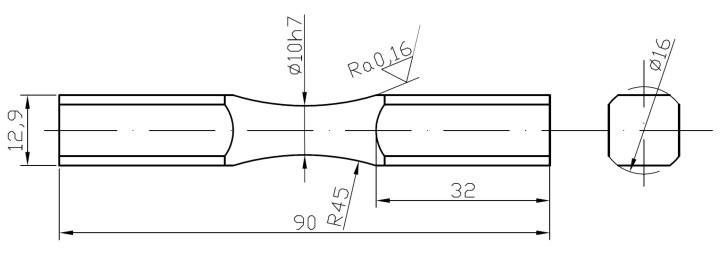
Shape and dimensions of fatigue test specimen, (dimensions are in mm).

**Figure 3 materials-13-04850-f003:**
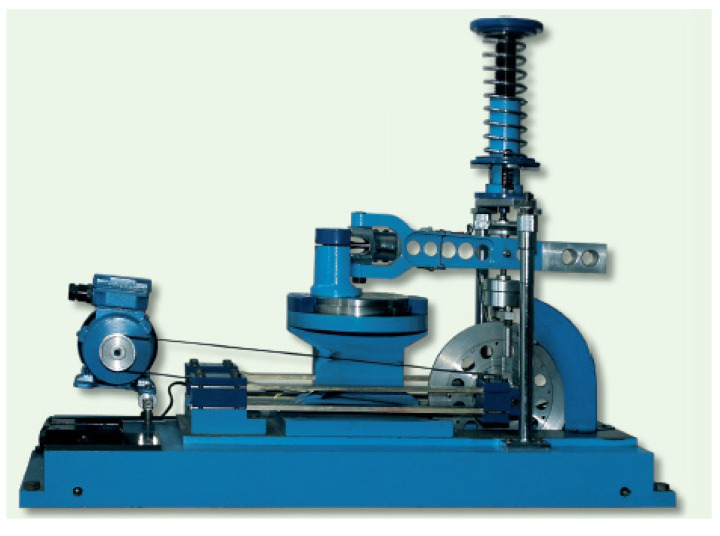
Fatigue test stand with controlled torque.

**Figure 4 materials-13-04850-f004:**
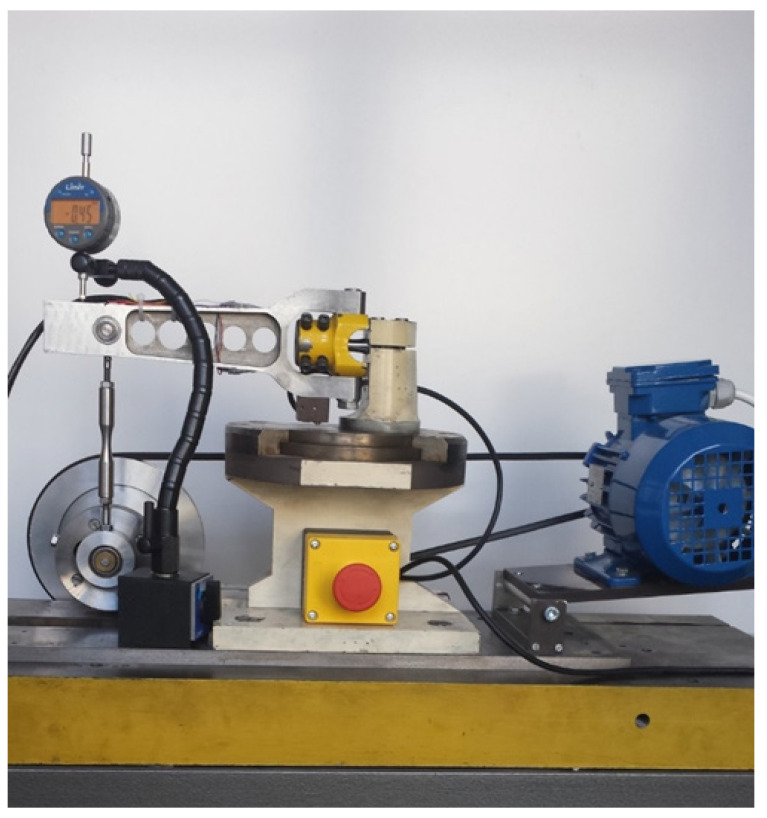
Fatigue test stand with controlled strain.

**Figure 5 materials-13-04850-f005:**
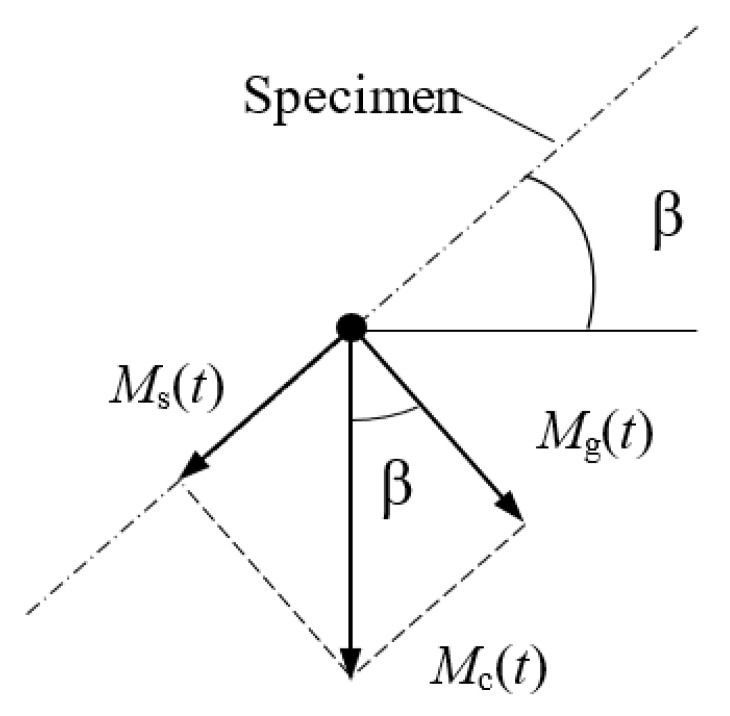
Interpretation of lever angle β.

**Figure 6 materials-13-04850-f006:**
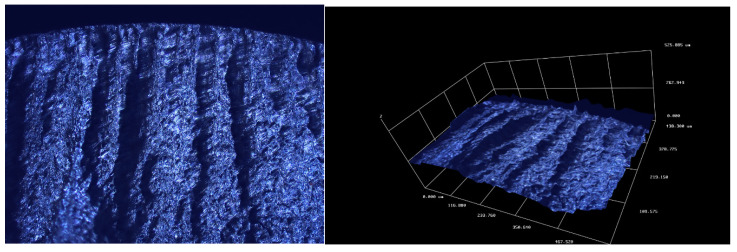
Microscopic image of the fatigue break obtained for bending of the aluminum specimen 6082-T6.

**Figure 7 materials-13-04850-f007:**
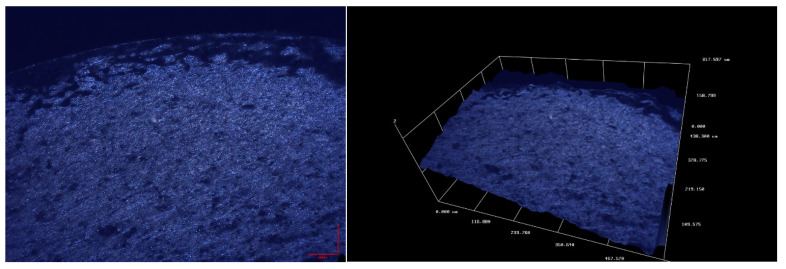
Microscopic image of the fatigue break obtained for torsion of the aluminum specimen 6082-T6.

**Figure 8 materials-13-04850-f008:**
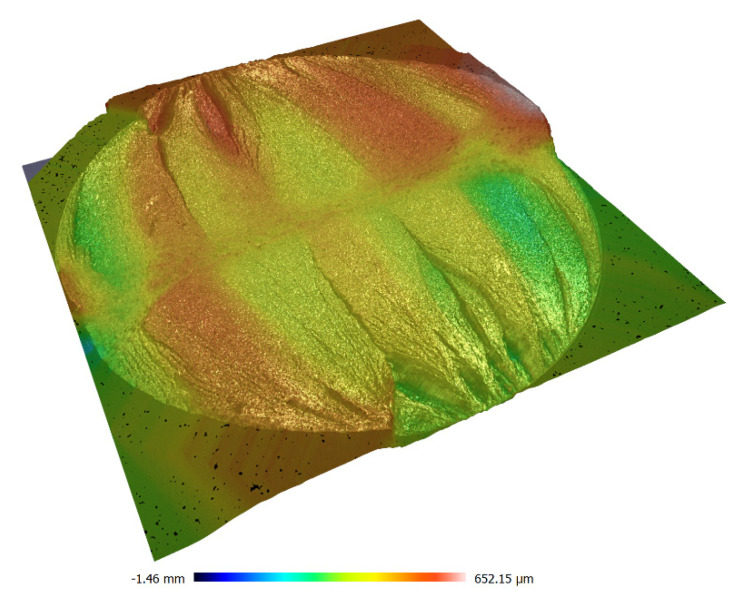
Macroscopic image of the fatigue break obtained for bending of the aluminum specimen 6082-T6.

**Figure 9 materials-13-04850-f009:**
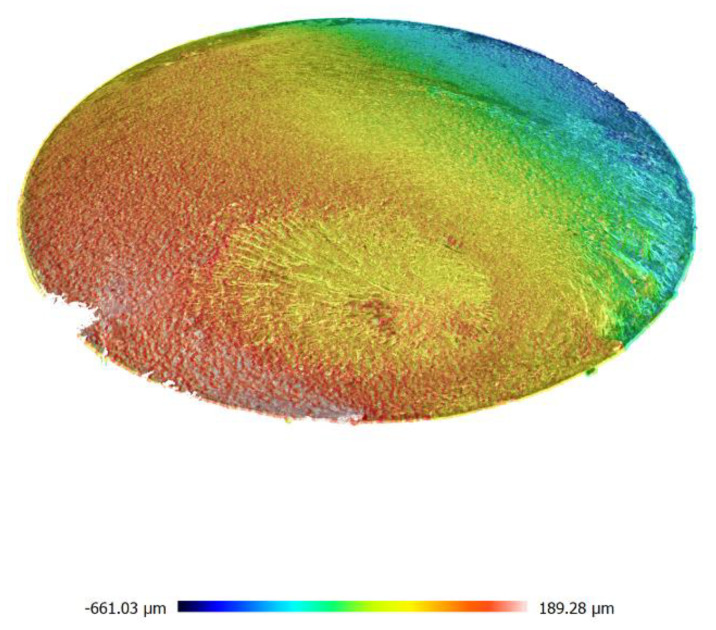
Macroscopic image of the fatigue break obtained for torsion of the aluminum specimen 6082-T6.

**Figure 10 materials-13-04850-f010:**
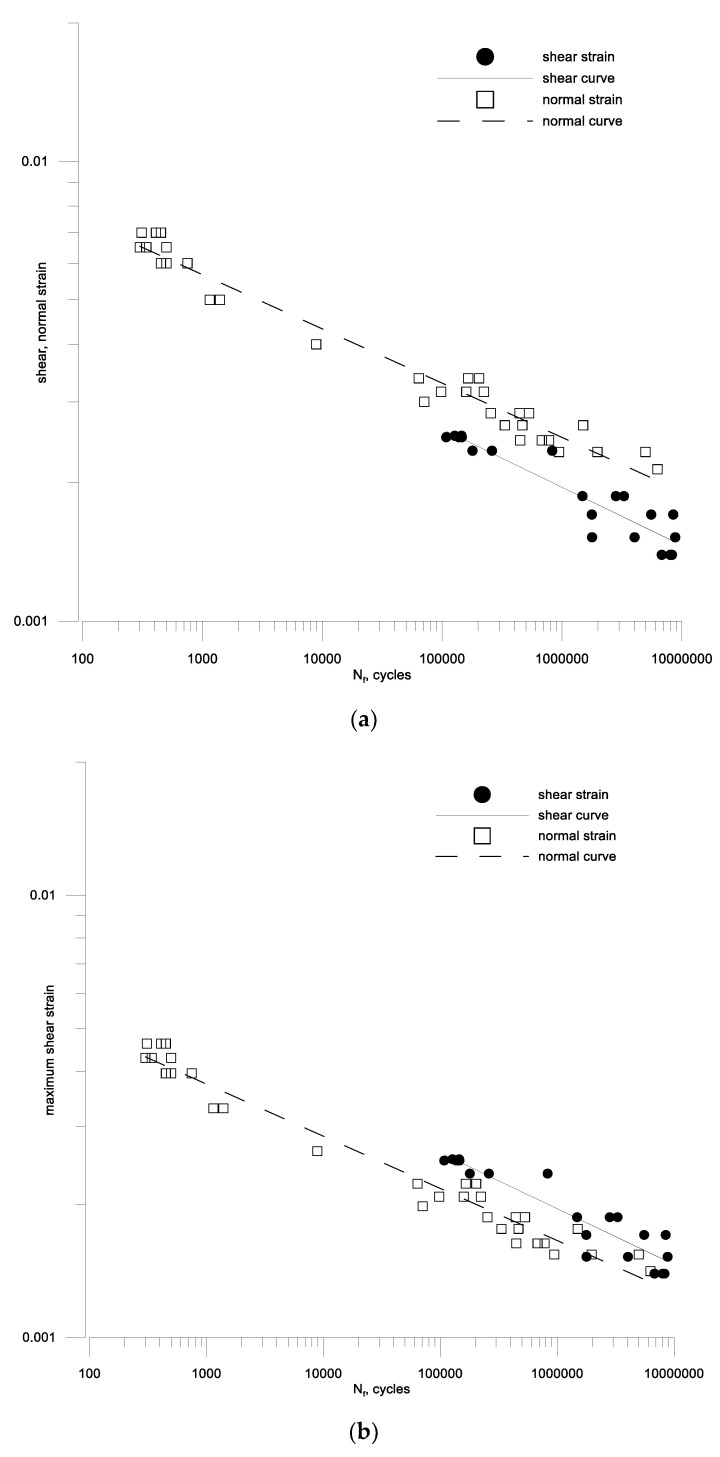
Fatigue characteristics for oscillatory bending and symmetrical torsion (**a**) according to the set strain amplitudes (**b**) according to the maximum shear strain amplitude for bending and torsion of aluminum 2017A-T4.

**Figure 11 materials-13-04850-f011:**
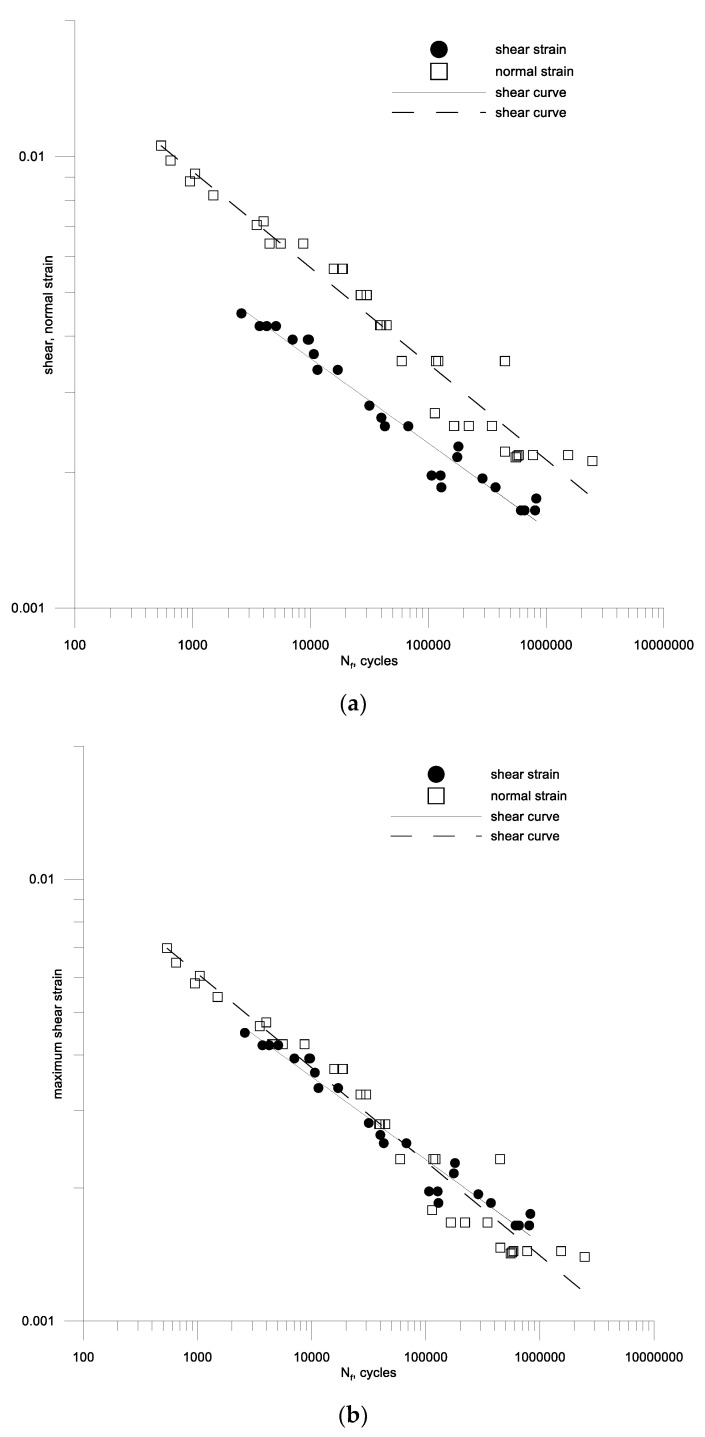
Fatigue characteristics for oscillatory bending and symmetrical torsion (**a**) according to the set strain amplitudes (**b**) according to the maximum shear strain amplitude for bending and torsion of aluminum 6082-T6.

**Figure 12 materials-13-04850-f012:**
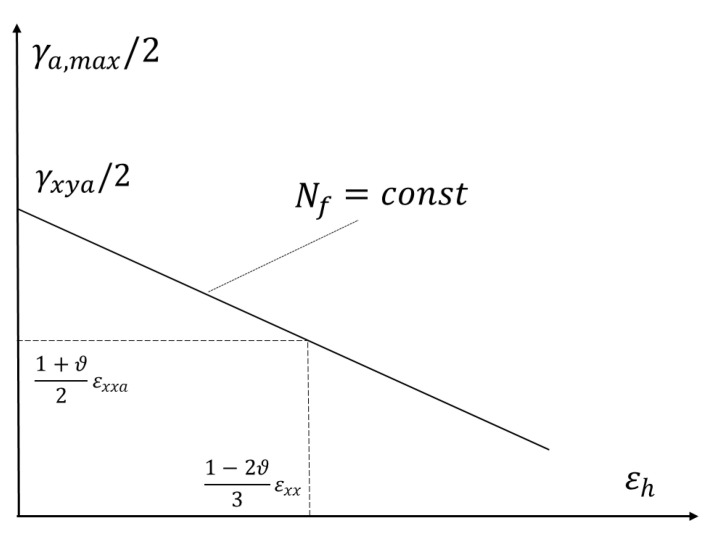
Graphical interpretation takes the amplitude of the maximum shear and hydrostatic strain into consideration.

**Figure 13 materials-13-04850-f013:**
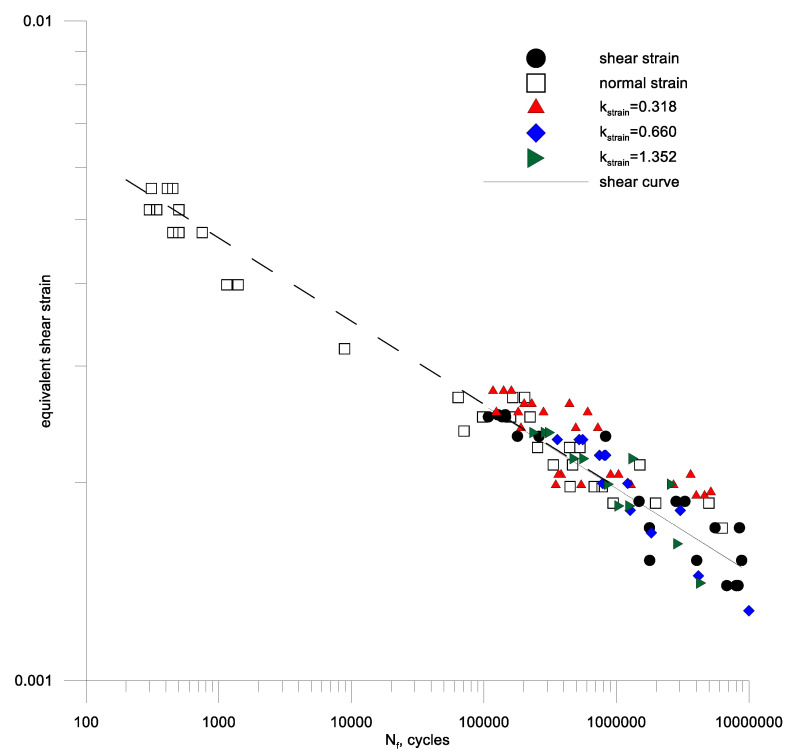
Equivalent strain in the plane of the maximum shear strain including hydrostatic strain for bending and torsion combinations of aluminum samples 2017A-T4.

**Figure 14 materials-13-04850-f014:**
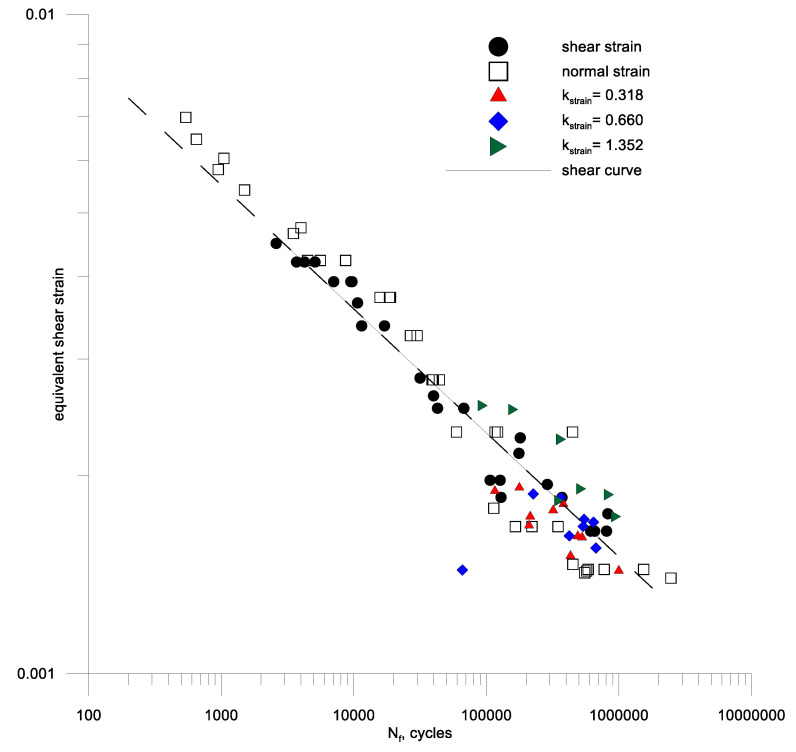
Equivalent strain in the plane of the maximum shear strain including hydrostatic strain for bending and torsion combinations of aluminum samples 6082-T6.

**Table 1 materials-13-04850-t001:** Chemical composition of aluminum alloys (in %) (Al—the rest).

Aluminum	Cu	Mg	Mn	Si	Fe	Zr + Ti	Zn	Cr
2017(A)	3.5–4.5	0.4–1.0	0.4–1.0	0.6	<0.7	<0.25	<0.25	<0.25
6082	<0.1	0.6–1.2	0.4–1.0	0.7–1.3	<0.5	<0.1	<0.2	<0.25

**Table 2 materials-13-04850-t002:** Basic mechanical parameters of analyzed aluminum alloys.

Aluminum	E, GPa	σ_y_, MPa	σ_u_, MPa	A5 %	ν
2017(A)-T4	72	395	545	21	0.32
6082-T6	77	365	385	27.2	0.32

**Table 3 materials-13-04850-t003:** Cyclic parameters of analyzed aluminum alloys.

Aluminum	K’, MPa	n’	σ’f, MPa	ε’f	b	c
2017(A)-T4	617	0.066	643	1.879	−0.065	−0.988
6082-T6	616	0.099	533	0.185	−0.066	−0.634

**Table 4 materials-13-04850-t004:** Cyclic parameters of analyzed aluminum alloys.

Aluminum	ε’’_f_	b_ε_	γ’’_f_/2	b_γ_	$\frac{\gamma_{af}/2}{\varepsilon_{af}}(10^{5})$	$a_{\varepsilon }(10^{5})$
2017A-T4	0.0128	−0.118	0.0112	−0.126	1.208	1.14
6082-T6	0.0403	−0.213	0.0202	−0.188	0.968	−0.18

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
