# Peer review of "Fatigue Life of Aluminum Alloys Based on Shear and Hydrostatic Strain"

_materials, 2020, doi:10.3390/ma13214850_

Round 1
Reviewer 1 Report
Dear Authors,
The aim of this study was to propose fatigue strain criterion to evaluate fatigue life of aluminum alloys. While the topic is fitting to the journal scope, The aim of this study was to propose fatigue strain criterion to evaluate fatigue life of aluminum alloys. Fatigue life prediction is important research target and would be useful information for readers in Materials. Revise the manuscript by following comments.
Major points
There was no clear purpose in the Abstract section and the Introduction section. Add the clear purpose of this study. To propose new fatigue strain criterion? or to evaluate fatigue life of aluminum alloys?
Manuscript format (Introduction, Materials and Methods, Results, Discussion, and Conclusion) should be revised. Current Discussion and Conclusion section is not containing Discussion. Methods and Results should be separated.
Minor points
There is no information about details of microscopy. Add them.
Figure legends of Figure 3 and Figure 4 were same. What's the difference between two figures?
Table 1
"0.6/1.2" is typo? It should be modified to "0.6-1.2". Make sure all the related points and revise them.
Table 2
"Rm ,MPa" should be modified to "Rm, MPa".
Equation 5
Which is appropriate between epsilon_a, p or epsilon_ap? Make sure all the related points and revise them.
Page 3, Line 93 and 94
"a, t", "a, e", and "a, p" should be described in lower case. They seem in middle case. Revise them.
Figure 1
What is the unit? Add the appropriate unit. Same for Figure 2.
What's the meaning of gray photos in Figure 8 and Figure 9? Color photos are enough to understand?
Author Response
Thank you for your review.
We hope that after your comments and suggestion our revised manuscript is easier to read and understand, also in general more useful for the reader.
See attched file

Reviewer 2 Report
The manuscript is devoted to the definition of new criteria for fatigue fracture of aluminum alloys.
The work is of interest for the scientific community, but the presentation of the results obtained is not optimal. The manuscript is difficult to read, there is no clarity in the presentation of the material.
The manuscript requires significant improvement in design, consistency of data presentation.
1) Lines 32-33. Transfer the data on the equipment used to section 3 "Experimental research".
2) Lines 30-32. The purpose of the work should be stated at the end of the "Introduction" section and should be more clearly formulated.
3) Lines 221-238. The literature review should be moved to the "Introduction" section.
4) The "Introduction" lacks literature data on the mechanical characteristics of the investigated aluminum alloys. The alloys are quite common; these data are available in the literature and should be presented.
5) Consider renaming section 2. Maybe "Fatigue Fracture Criteria"? Then Lines 39-69 can be moved to the second section, which will help the clarity and consistency of the presentation of the material.
6) Fig. 1a, 10 and 11 are of poor quality.
7) Consider transferring Fig. 1-4 in Supplementary Materials. These figures do not carry a large scientific load and characterize technical aspects of mechanical testing.
8) It is necessary to bring photographs of the fractures of the samples from a scanning electron microscope (without coloring). That is, to carry out a fractographic analysis describing the nature (type) of the resulting fractures.
After making these changes, it will be possible to read the manuscript as a whole material and delve into the scientific aspects of the results.
Author Response
Thank you for your review. We hope that after your comments and suggestion our revised manuscript is easier to read and understand, also in general more useful for the reader.
See atached file:

Round 2
Reviewer 1 Report
Dear Authors, The manuscript was well revised by following the reviewer's comments.Author Response
Thank you for review our paper. Now, The paper is more usefull for readers
Reviewer 2 Report
The authors did a lot of work to improve the presented manuscript, but:
If we are talking about fracture and their characteristics, then it is necessary to devote one or two paragraphs in the manuscript to fractographic research. It is necessary to describe the type of fractures after testing, whether it is the same for two different grades of aluminum alloys, to estimate the average size of the facets, to provide one or two photographs for each of the samples (possibly in additional materials, and not in the main text of the manuscript), and
to correlate the obtained results on mechanical characteristics with fractographic analysis.
Author Response
Thank You for this comment. It is fact very true that in order to properly discuss the nature of those fractures deeper fractographic analysis should be done. On the other hand this paper was designed and written to propose a new criterion, and to verify it on the basis of fatigue tests. We chose two aluminium alloys. Neither the criterion or the paper itself requires fracture type information to properly calculate fatigue life. We only use fatigue life curves, and basic experimental data in general.
None of the authors is a specialist in metallography and even less in fractography. Therefore we decided not to pursue this part of fatigue related filed.
We also believe that fracture analysis would add very little to the paper as it is mainly about the proposal of new criterion. Criterion which doesn’t need any of that information to work properly.
Once again thank You for this suggestion but we would rather not put any more information about the specimens fractures into this paper.